

# GP4ESP: a hybrid genetic algorithm and particle swarm optimization algorithm for edge server placement

Fang Han[1], Hui Fu[1], Bo Wang[2], Yaoli Xu[2] and Bin Lv[3]

[1] Huanghe Science and Technology University, Zhengzhou, China
[2] Software Engineering College, Zhengzhou University of Light Industry, Zhengzhou, China
[3] Linyi Vocational University of Science and Technology, Linyi, China

## ABSTRACT

Edge computing has attracted wide attention due to its ultra-low latency services, as well as the prevalence of smart devices and intelligent applications. Edge server placement (ESP) is one of the key issues needed to be addressed for effective and efficient request processing, by deciding which edge stations to equip with limited edge resources. Due to NP-hardness of ESP, some works have designed meta-heuristic algorithms for solving it. While these algorithms either exploited only one kind of meta-heuristic search strategies or separately perform two different meta-heuristic algorithms. This can result in limit performance of ESP solutions due to the "No Free Lunch" theorem. In addition, existing algorithms ignored the computing delay of edge servers (ESs) on request process, resulting in overestimation of the service quality. To address these issues, in this article, we first formulate ESP problem with the objective of minimizing the overall response time, considering heterogeneous edge servers with various service capacity. Then, to search effective or even the best ESP solutions, we propose a hybrid meta-heuristic algorithm (named GP4ESP) by taking advantage of both the powerful global search ability of genetic algorithm (GA) and the fast convergence of particle swarm optimization (PSO). GP4ESP effectively fuses the merits of GA and PS by integrating the swarm cognition of PSO into the evolutionary strategy of GA. At last, we conducted extensive simulation experiments to evaluate the performance of GP4ESP, and results show that GP4ESP achieves 18.2%–20.7% shorter overall response time, compared with eleven up-to-date ESP solving algorithms, and the performance improvement is stable as the scale of ESP is varied.

# INTRODUCTION

In the present age, smart devices are increasingly prevalent in all aspects of our lives. Such as quite a lot of people always has their smartphones in hands. The number of connected devices will reach 75 billion by 2025 (*Kiruthiga Devi & Padma Priya, 2023*). On the other hand, there is an increasing variety of services assessed by various devices everywhere and anytime, as the development of information and communication technology. Due

Corresponding author
Bo Wang, wangb@zzuli.edu.cn

to diversity of users, centralised cloud computing can no longer meet their demands, especially for providing low latency services. Therefore, edge computing has attracted wide attention as it can provide ultra-low latency services (*Dayong et al., 2024*).

Edge computing is deploying several edge stations nearby end users. Each station is equipped with an access point (AP), and some stations with an edge server (ES). Users can access services provided by edge computing over neighbouring access points, *e.g.*, micro base station and wireless router, and their requests are processed by ESs or the centralized cloud when ESs have inadequate capacities. Due to the limited budget and cost-effectiveness, there are not enough ESs for all edge stations. In addition, there is very limited distance for an AP signal, and an edge station can communicate with users not beyond its AP's maximal signal distance. Thus, the edge server placement (ESP) heavily influences users' request processing performance and is one of the key issues needed to be addressed to decide which edge stations to equip with limited ESs for good cost performance (*Asghari & Sohrabi, 2024*).

Given a set of ESs and candidate positions (edge stations), ESP is to decide where each ES is placed, which plays a big part in service quality and resource efficiency. This is because unreasonable ESP solution can lead to poor response time for part of requests while underutilizations of some ESs at the same time. Due to NP-Hard complexity of ESP (*Asghari & Sohrabi, 2024*), there are mainly two kinds of methods for solving it, heuristic and meta-heuristic algorithms. Heuristic algorithms use local search strategies, which generally produce local optimal solutions with very little overhead. Meta-heuristic algorithms exploit global search strategies inspired by nature or/and society laws, which can provide better solutions than heuristic algorithms but have more overhead. ESP is not time-sensitive as its solutions usually required by service providers for building or upgrading their edge computing platforms every few months or even years. Thus, in this article, we focus on meta-heuristic algorithms for finding good or even the best solutions of ESP.

At present, most of existed ESP solving algorithms considered exploiting only one kind of meta-heuristic search strategies, *e.g.*, particle swarm optimization (PSO) (*Tiwari et al., 2024*; *Pandey et al., 2023*), Artificial Bee Colony (ABC) (*Zhou, Lu & Zhang, 2023*), Whale Optimization Algorithm (WOA) (*Moorthy, Arikumar & Prathiba, 2023*), *etc.* These algorithms provide ESP solutions with limit performance, because of the "No Free Lunch" theorem (*Adam et al., 2019*) that no single algorithm performs well all the time. By combining advantages of two or more algorithms, hybrid meta-heuristic algorithms can have more powerful global search abilities, and thus achieve better solutions. There are several existing works focusing on designed hybrid algorithms for ESP (*Ma, 2021*; *Bahrami, Khayyambashi & Mirjalili, 2024*; *Asghari, Sayadi & Azgomi, 2023*; *Asghari, Azgomi & Darvishmofarahi, 2023*). But they only separately perform different meta-heuristic algorithms, resulting in inefficient hybridization. In addition, these existing works chose meta-heuristic algorithms at random for hybridizing, without considering the complementarity of different kinds of algorithms. These lead to an inefficient hybridizing schemes. In addition, existing works are only concerned about the network transmission

delay for evaluating the performance of requests, ignoring the computing delay on ESs and thus leading to overestimation of the service quality for ESP solutions.

In this article, to address the above issues for effectively solving ESP, we design a hybrid global search strategy with ideas of GA and PSO. These two meta-heuristic algorithms are both the most representative meta-heuristic algorithms, and they have a strong advantage mutual-complementarity because GA has powerful global search ability but slow convergence (*Katoch, Chauhan & Kumar, 2021*) while PSO has fast convergence but is easily trapped into local optima (*Nayak et al., 2023*). The contributions of this article can be summarized briefly as follows.

- ESP is formulated into a binary linear programming problem in a universal three-layer device-edge-cloud computing framework, to decide which edge stations for placing fixed number of edge servers with minimized average response time of all user requests.
- A hybrid meta-heuristic algorithm is designed for solving ESP, with the search strategy combining evolutionary operators of GA and swarm cognition of PSO.
- Extensive experiments are conducted for evaluating the performance of the designed hybrid algorithm in effectiveness and efficiency as well as performance stability with varied problem scale.

In the remainder of this article, we discuss related work in the second section. We formulate ESP in the third section, and illustrates the proposed meta-heuristic algorithm in the forth section. In the fifth section, we present the experiment results. And at last, we conclude our work in the sixth section.

## RELATED WORK

As the increasing attention drawn to edge computing, a great deal of work focused on improving the resource efficiency and the service quality of edge services in various aspects, *e.g.*, edge station deployment (*Xing, Song & Wang, 2023*), edge server placement (*Bahrami, Khayyambashi & Mirjalili, 2023*; *Asghari & Sohrabi, 2024*), edge service caching (*Barrios & Kumar, 2023*) and edge task offloading (*Wang et al., 2023*). In this article, we focus on the ESP problem to optimize the performance of request processing by placing purchased edge servers on deployed edge stations. Table 1 summarizes the characteristics of related works.

Due to the NP-hardness of ESP problem, most of the works exploited meta-heuristic algorithms to solve it. *Tiwari et al. (2024)* first transformed the ESP problem into a 0-1 Knapsack problem, and then proposed to use PSO for solving the Knapsack problem. *Zhou, Lu & Zhang (2023)* used ABC to determine the location of edge servers and distributes requests between ES for load balance. By comparing with K-means clustering method, ABC achieves better load balance. This work didn't consider distributing some requests to the cloud, which can result in a poor performance when the system load is high. *Chen et al. (2023)* used the Non-dominated Sorting Genetic Algorithm (NSGA) for solving ESP problem to improve the average network delay and the processing delay. *Pandey et al. (2023)* analysed the performance of three methods (PSO, Top-First and Random) on ESP in busy hours and non-busy hours, respectively, and results show PSO achieves the best performance in improving resource utilization, energy consumption and the number of

**Table 1  Comparison of our proposed method with related works.**

| Work | Objective | Edge server | Algorithm |
|------|-----------|-------------|-----------|
| *Tiwari et al. (2024)* | load balance and energy | heterogeneous | PSO |
| *Zhou, Lu & Zhang (2023)* | delay | homogeneous | ABC |
| *Chen et al. (2023)* | delay and queue length | heterogeneous | NSGA |
| *Pandey et al. (2023)* | load balance and energy | homogeneous | PSO |
| *Moorthy, Arikumar & Prathiba (2023)* | latency and energy | homogeneous | WOA |
| *Zhang et al. (2023)* | response time | homogeneous | PSO |
| *Ma (2021)* | delay, energy and load balance | heterogeneous | GA and PSO sequentially |
| *Bahrami, Khayyambashi & Mirjalili (2023)* | delay, coverage and cost | homogeneous | NSGA-II and MOPSO separately |
| *Asghari, Sayadi & Azgomi (2023)* | delay and energy | heterogeneous | BOA and CRO hierarchically |
| *Asghari, Azgomi & Darvishmofarahi (2023)* | delay and energy | heterogeneous | WOA and Game hierarchically |
| This article | response time | heterogeneous | hybrid GA and PSO |

ESs required for meeting requirements. *Moorthy, Arikumar & Prathiba (2023)* designed a WOA-based algorithm for solving ESP, and achieved better performance than PSO by conducted experiment results. While, their experiments are small-scale, where the number of ESs is at most 30, which leads to unknown efficiency and effectiveness of their proposed algorithm in large-scale ESP. *Zhang et al. (2023)* exploited niching technology to improve PSO on solving ESP as ESP is generally multimodal optimization problem. This work grouped particles according to their Jaccard similarity coefficients at first in each iteration, and replaced the global best position with the niche best ones in updating each particle solution.

All of the above works exploited only one kind of meta-heuristic algorithms, resulting in limited search ability. Thus, *Ma (2021)* proposed an algorithm trying to combine GA and PSO. While, this work was just sequentially performing GA and PSO on the whole population. *Bahrami, Khayyambashi & Mirjalili (2024)* considered a multiple objective ESP for minimizing latency, maximizing coverage and minimizing the cost of rented ESs. They proposed to combine NSGA-II and Multiple Objective PSO (MOPSO) to solve the ESP. In this work, the authors conducted NSGA-II and MOPSO on upgrading the two halves of the whole population, respectively, in each evolutionary iterative time. *Asghari, Sayadi & Azgomi (2023)* and *Asghari, Azgomi & Darvishmofarahi (2023)* first divided the physical space that the ESP is concerned about into several small zones to decrease the problem complexity. Then, these two works respectively exploited Butterfly Optimization Algorithm (BOA) and WOA to search the global optimal solution in the whole space, and Coral Reefs Optimization (CRO) algorithm and game theory to make the local optimal decision for each zone. All of these hybridization approaches are conducting different optimizers separately for solution searching, which cannot effectively fused advantages of both algorithms.

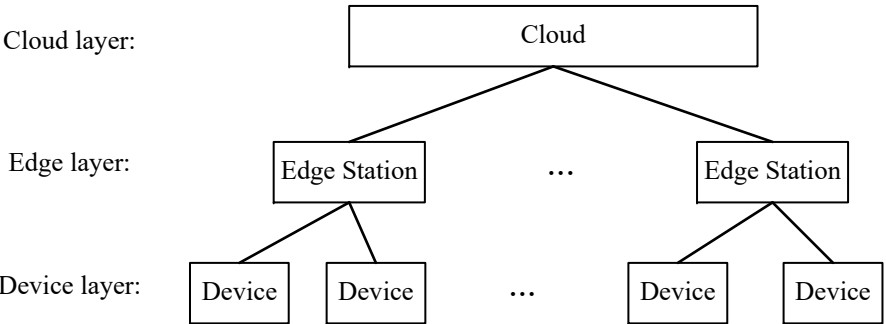

Cloud layer:

Edge layer:

Device layer:

**Figure 1  The three-layer device-edge-cloud computing framework.**

All of these above existing works exploited either only one kind of meta-heuristic algorithms or separately two different meta-heuristic algorithms for solving ESP. This results in that they hardly effectively fuse advantages of different meta-heuristic algorithms, leading to limited performance on the global solution search. To overcome these issues, in this article, we designed a hybrid meta-heuristic algorithm to fuse the advanced search ideas of GA and PSO for solving ESP with heterogeneous ESs that have various service capacities.

## PROBLEM FORMULATION

In this article, we consider a universal three-layer device-edge-cloud computing framework, as shown in Fig. 1, composed of the device, edge, and cloud layers. Users initiate their requests by various devices of the device layer. These requests are directed to their neighbouring edge station(s) of the edge layer for their processing. When an edge station has not enough capacity for serving all received requests, it will redirect some requests to the cloud layer to extend its service capacity. Base on the three-layer framework, we will formulate ESP problem below. The notations used for our formulation are summarized in Table 2

Considering an ESP situation, the service provider decides to build or upgrade its edge computing due to the business growth by purchasing a set of ESs that is represented by $e_1, e_2, \ldots, e_E$. For $j$th ES, $e_j$, its service capacity is $\mu_j$, that is to say, $e_j$ can process $\mu_j$ requests per unit time on average. There have been $S$ edge stations deployed distributively, represented as $s_1, s_2, \ldots, s_S$. The maximum signal distance of the AP on $s_i$ is $D_i$. The position of $s_i$ is $(x_i, y_i)$ that are horizontal and vertical values in orthogonal plane coordinate system or latitude and longitude in geographic coordinate.

In the edge service platform, there are $U$ users initiating requests by their respective devices. On average, $\lambda_k$ requests are initiated by the $k$th user, $u_k$, per unit time, and to be processed by the edge computing. The location of $u_k$ is $(l_k, m_k)$ that are coordinate values same to the station position. Then, the distance between every user and each edge station can be calculated by Eq. (1) when using orthogonal plane coordinate system, or Eq. (2)

**Table 2** The three-layer device-edge-cloud computing framework.

| Notation | Description |
| --- | --- |
| $S$ | Number of edge stations |
| $E$ | Number of purchased edge servers |
| $U$ | Number of users initiating requests |
| $s_i$ | $i$th edge station |
| $D_i$ | Maximum signal distance of AP in $s_i$ |
| $(x_k, y_k)$ | coordinate values of $s_i$ |
| $e_j$ | $j$th purchased edge server |
| $\mu_j$ | Service rate of $e_j$ for request processing |
| $u_k$ | $k$th user |
| $(l_k, m_k)$ | coordinate values of $k$th user's location |
| $\lambda_k$ | Number of requests initiated by $u_k$ per unit time |
| $d_{i,k}$ | Distance between $s_i$ and $u_k$ |
| $c_{i,k}$ | Indicator whether requests of $u_k$ can be received by $s_i$ |
| $r_i$ | Average response time of $s_i$ in processing received requests |
| $t_i$ | Average response time of requests received by $s_i$ and processed by edge and cloud collaboration |
| $T$ | Average response time of all requests |
| $z_{i,j}$ | Decision variable indicating whether $e_j$ is placed on $s_i$ |
| $R$ | Earth's mean radius |

approximately if employing geographic coordinate, where $R$ is the earth's mean radius.

$$d_{i,k} = \sqrt{(x_i - l_k)^2 + (y_i - m_k)^2} \tag{1}$$

$$d_{i,k} = R * \arccos(\cos l_k * \cos x_i * \cos(m_k - y_i) + \sin l_k * \sin x_i). \tag{2}$$

During the operator, a request can be received by the station that covers its user, *i.e.,* the distance from the user to the station is not exceeding the maximum signal distance of the station's AP. We use $c_{i,k}$ to represent whether a user's requests can be received by a station, which is defined in Eq. (3). Then, we can get the request rate on each edge station by accumulating request rates of all users that are covered by the station, that is $\sum_{k=1}^{U}(c_{i,k} * \lambda_k)$ for $s_i$.

$$c_{i,k} = \begin{cases} 1, & \text{if } d_{i,k} \leq D_i \\ 0, & \text{if } d_{i,k} > D_i. \end{cases} \tag{3}$$

For formulating ESP, we define binary variables $z_{i,j}$ $(i = 1, 2, \ldots, S; j = 1, 2, \ldots, E)$ to represent the ESP solution, where $z_{i,j}$ is 1 if $e_j$ is placed on $s_i$ and 0 if not. Then, given an ESP solution, the service rate of $s_i$ is $\sum_{j=1}^{E}(z_{i,j} * \mu_k)$, and we can achieve the average response time of requests received and processed by the station by Eq. (4) based on M/M/1 queue model in queue theory.

$$r_i = \frac{1}{\sum_{j=1}^{E}(z_{i,j} * \mu_k) - \sum_{k=1}^{U}(c_{i,k} * \lambda_k)}. \tag{4}$$

When the load, the request rate, is too high for an edge station, the average response time can be longer than requests processed by the cloud that is assumed as a constant ($\tau$) as it generally has abundant resources. In such case, we can offload a portion of requests to the cloud for improving the overall average response time. At worst, all requests can be offloaded to the cloud, which produces an average response time of $\tau$. Thus, by considering the collaboration of edge and cloud computing, the average response time of requests received by $s_i$ can be improved from Eq. (4) to Eq. (5). And the overall average response time of requests in the edge service system can be achieved by Eq. (6), where the denominator is the accumulated number of requests per unit time, and the numerator is the accumulated response time of all requests per unit time.

$$t_i = \begin{cases} r_i, & \text{if } 0 < r_i < \tau \\ \tau, & \text{else} \end{cases} \tag{5}$$

$$T = \frac{\sum_{i=1}^{S} (\sum_{k=1}^{U} (c_{i,k} * \lambda_k) * t_i)}{\sum_{k=1}^{U} \lambda_k}. \tag{6}$$

Based on the above formulations, ESP can be modelled by following optimization problem. The decision variables include $z_{i,j}$, $i = 1, 2, \ldots, S$; $j = 1, 2, \ldots, E$, that represent a ESP solution indicating the station where each ES is placed. The optimization objective Eq. (7) is minimizing the overall average response time that is directly related to the service quality, different from other works that use the network delay as the performance indicator. Constraints Eq. (8) are calculating the overall average response time given a solution by queuing theory. Constraints Eq. (9) indicate the atomicity of ESs, that every ES cannot be placed on two or more stations. Constraints Eq. (10) restrict the possible values of decision variables that are all binary.

minimizing $T$, $\tag{7}$

subject to,

Eq. (1)–(6) $\tag{8}$

$$\sum_{i=1}^{S} z_{i,j} \leq 1, j = 1, 2, \ldots, E, \tag{9}$$

$$z_{i,j} \in \{0, 1\}, i = 1, 2, \ldots, S, j = 1, 2, \ldots, E. \tag{10}$$

The ESP problem has been proven as NP-hard (*Bahrami, Khayyambashi & Mirjalili, 2023*), thus no existing method can exactly solve the problem with middle to large scales by reasonable time. Therefore, in the next section, we design a hybrid meta-heuristic algorithm to search a good or even the global best solution for ESP with polynomial time.

## HYBRID META-HEURISTIC EDGE SERVER PLACEMENT

GA and PSO are both the most widely used and representative meta-heuristic algorithms (*Katoch, Chauhan & Kumar, 2021*; *Nayak et al., 2023*). GA is imitating the natural evolutionary law to design a global search strategy with crossover, mutation, and selection operators (*Katoch, Chauhan & Kumar, 2021*; *Alhijawi & Awajan, 2023*). Benefits from these three operators, GA achieves a population with high diversity, resulting in a powerful global search ability. While, every coin has its two sides, and GA is no exception. GA generally has a slow convergence rate particularly in solving large scale problems. PSO is designed based on rules of flock feeding, which moves each individual (particle) toward the local and global best position with self- and social cognitions. Contrary to GA, PSO has a quick convergence but is easily falling into local positions. Thus, GA and PSO have complementary advantage, and we consider to effectively fuse their global search ideas for solving ESP in this section. The framework of the hybrid GA and PSO for ESP is detailed as follows, named as GP4ESP (hybrid GA and PSO for ESP) in this article, as shown in Fig. 2 and Algorithm 1.

---

**Algorithm 1** GP4ESP: a hybrid GA and PSO for ESP

---

1: initializing a population randomly;
2: evaluating the fitness for every individual of the initialized population;
3: recording the personal best as the current code for each individual;
4: recording the global best as the individual with the best fitness;
5: **while** the maximum iterative number is not reached **do**
6:     **for** each individual **do**
7:         crossing the individual with another random individual; //original GA
8:         crossing the individual with its personal best; //self-cognition of PSO
9:         crossing the individual with the global best; //social-cognition of PSO
10:         evaluating the fitness of offspring produced by lines 7–9;
11:         updating the individual as the offspring with the best fitness;
12:         updating the personal best as the individual if the latter has better fitness;
13:         updating the global best as the individual if the latter has better fitness;
14:         mutating the individual;
15:         evaluating the fitness of the individual;
16:         updating the personal best as the individual if the latter has better fitness;
17:         updating the global best as the individual if the latter has better fitness;
18: **return** decoded global best chromosome;

---

At first, we should build a solution space, *i.e.,* design a solution encoding/decoding method, for GP4ESP's searching according to features of the ESP problem. The solution space should include all available ESP solutions, with a one-to-one correspondence from individuals (chromosomes in GA and particle positions in PSO) to ESP solutions generally. In this article, due to the one-to-many relationship between edge stations and ESs, *i.e.,*

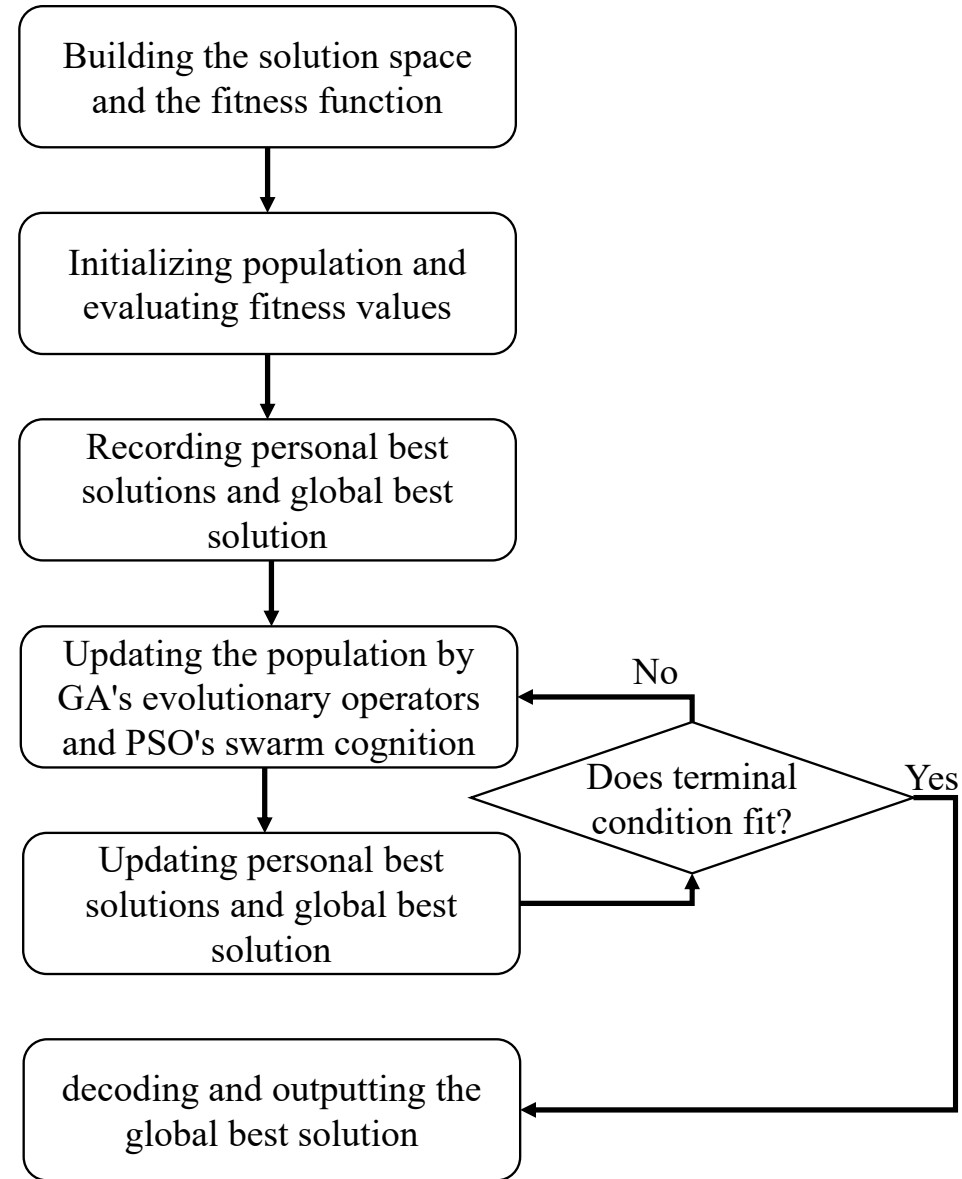

**Figure 2** The framework of hybrid meta-heuristic algorithm for edge server placement.

each ES can be placed on at most one edge station while every edge station can be equipped with two or more ESs, GP4ESP employs an integer coding approach for the map between individuals and ESP solutions.

In the employed integer coding approach, each individual in the solution space is represented as a vector with $E$ dimensions corresponding to $E$ ESs. The value in each dimension represents the NO. of the edge station where the corresponding ES is placed, which ranges from 1 to $S$, *i.e.,* Eq. (11) holds for ES $e_j$, where $v_j$ is value of an individual in the $j$th dimension. By such coding, the atomicity constraints Eqs. (9) and (10) is ensured, which is very helpful for avoiding to search unavailable solutions caused by constraint

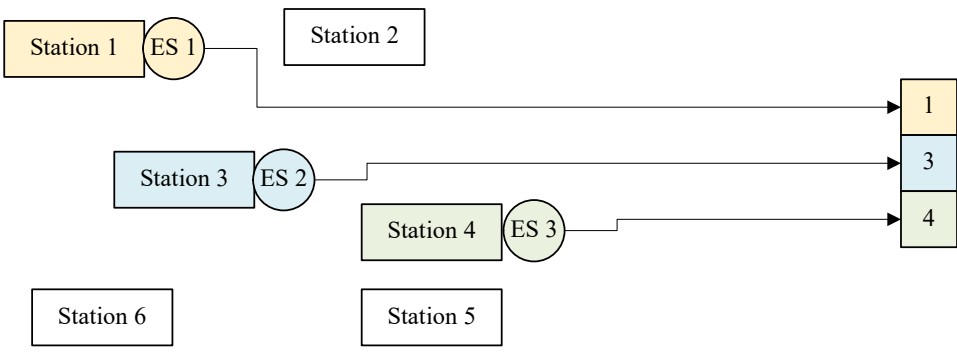

An ESP solution                                              The coded solution

**Figure 3  An example illustrating the integer coding approach for ESP.**

violations and dramatically reducing the searching space compared with the binary coding approach.

$$v_j = i \Leftrightarrow z_{i,j} = 1 \wedge \forall i'(i' \neq i \Rightarrow z_{i',j} = 0). \tag{11}$$

For example, as shown in Fig. 3, if there are 3 ESs to be placed on 6 edge stations, an individual has 3 dimensions each of which has possible values from 1 to 6. The individual $\langle 1, 3, 4 \rangle$ represents that the three ESs are placed on the first, third and fourth edge stations, respectively.

Based on the built solution space, GP4ESP needs a fitness function to evaluate the goodness of individuals, which is the essential foundation for searching when employing meta-heuristic algorithms. In this article, GP4ESP uses the overall average response time $(T)$ of the corresponding ESP solution as the fitness of each individual. For calculating an individual's fitness value, GP4ESP first map the individual into the corresponding solution and achieves values of decision variables, $z_{i,j}$ $(i = 1, 2, \ldots, S; j = 1, 2, \ldots, E)$, by Eq. (11), in the optimization problem Eq. (7). Then, by Eq. (6), the value of $T$ can be easily got.

Given the solution space and the fitness function, GP4ESP initializes a population consisting of multiple individuals and evaluates fitness values of all individuals (lines 1 and 2). Then, GP4ESP records the personal best solution as the initialized one for each individual and the global one as the best individual with the best fitness (the minimum $T$). After these initializations, GP4ESP iteratively evolves the population and updates the personal and the global best solutions (lines 3 and 4), until the pre-set terminal condition is reached (line 5). There are generally two methods for setting the terminal condition. One is setting the maximum number of iterations, and another is the maximum iterative number that the best fitness is not improved (significantly). The population updating strategy exploited by GP4ESP is as follows.

For each individual, GP4ESP produces six offspring by combining the crossover operator and the swarm cognition (lines 7–9). GP4ESP conducts the crossover operator on each individual with a certain probability three times, where two offspring are produced in each

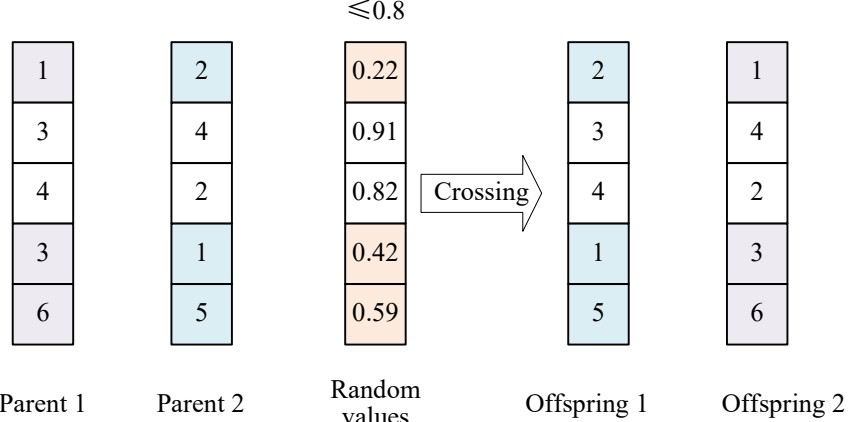

**Figure 4**  Example illustrating the crossover operator with the crossover probability of 0.8.

time. First, the individual is crossed with another individual, same to GA (line 7). Then, to exploit the self-cognition, the individual is crossed with its personal best solution (line 8). Finally, to take advantage of the social cognition, a crossover operator is conducted on the individual with the global best solution (line 9). For the produced six offspring, GP4ESP evaluates their fitness values, and update the individual as the best offspring (lines 10 and 11). After that, if the current individual has better fitness than its personal best solution, the personal best solution is updated as the current individual (line 12). And the global best solution is updated as the current individual if the global best solution has worse fitness (line 13).

To ensure the population diversity for exploiting the huge search spaces of large-scale ESP problems, GP4ESP employs the uniform crossover operator, as illustrated in Fig. 4. Given two selected individuals (parents), in each dimension, a random value between 0 and 1 is generated. If the generated random value is smaller than the pre-set crossover probability, the values of parents are exchanged in the corresponding dimension. After the same operation on every dimension, two new individuals (offspring) can be generated.

To increase the population diversity further for improving the global search ability, GP4ESP performs a mutation operator on each individual with a certain probability, same to GA (line 14). After the individual is mutated, its fitness is evaluated, and the personal and global best solutions are updated, respectively, if the mutated individual has better fitness (lines 15–17). In this article, we use the uniform mutation operator, as illustrated in Fig. 5, which changes the value of an individual (parent) in a dimension into another random available value if a random value is smaller than the defined mutating probability.

After the above population updating terminates, GP4ESP derives and outputs the best ESP solution from the global best solution (line 18).

## EXPERIMENTAL MEASUREMENT

In this section, we conduct extensive simulated experiments to evaluate the performance of GP4ESP. We first perform the performance evaluation in an ESP case with fixed system

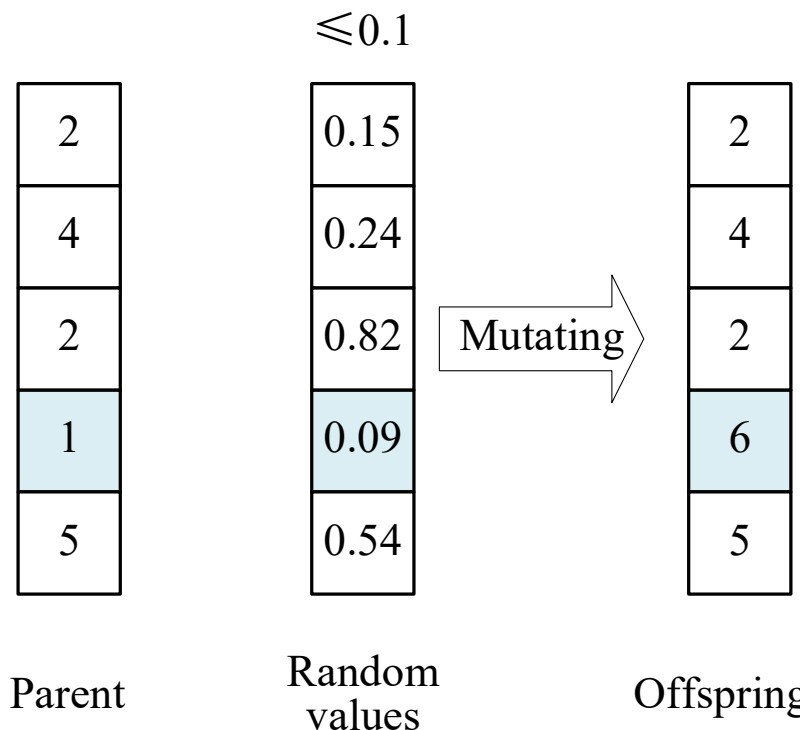

**Figure 5** Example illustrating the mutation operator with the mutating probability of 0.1.

**Table 3** The system parameter values in four cases.

|  | Station number ($S$) | ES number ($E$) | Load |
|---|---|---|---|
| Fixed | 1000 | 600 | 0.5 |
| Varied scale | 100, 200, …, 1000 | 0.6 * $S$ | 0.5 |
| Varied ES number | 1000 | 100, 200, …, 900 | 0.5 |
| Varied load | 1000 | 600 | 0.1, 0.2, …, 0.9 |

parameter values, and then, study the performance changes with varied parameters. In this article, we consider three system parameters that are the number of edge stations, the number of ESs and the load that is defined as the ratio of the total arrival rate on an edge station to an ES's service rate on average. We evaluate the performance variation in three experiment cases of varied scale, varied ES number and varied load. In the varied scale case, the numbers of edge stations and ESs are both changed. The system parameter values are shown in Table 3.

For the first system case with fixed parameter values, the numbers of edge stations and ESs are set as 1,000 and 60, respectively. The load is set as 0.5, where the service rate of each ES is randomly set in the range between 0 and 1,000, and the total arrival rate of every edge station is in the range of 0 to 500 (1,000 * 0.5). The response time of requests offloaded to the cloud is 50 ms (ms). For the case with varied problem scale, the number of stations is changed from 100 to 1,000, and the ES number is 0.6 times the station number. For the

**Table 4 The parameters set for these meta-heuristic algorithms.**

| Method | Parameter: Value |
|---|---|
| All | maximum iterative time: 100; population size: 100 |
| GA, DE, PSOM, GP4ESP | crossover probability: 0.8; mutating probability: 0.1 |
| SA | initial temperature: 100; termination temperature: 0.01; termination error: $1 \times 10^{-4}$; cooling coefficient: 0.99; annealing rate: 0.01; inner iterative time: 100 |
| PSO, PSOM, PSOSA, nPSO | inertia weight: linearly decreasing from 1.2 to 0.4; acceleration factors: 2.0 |
| ABC | threshold for scout update: 5 |

varied ES number case, the ES number is changed from 100 to 900. For the varied load case, the load is changed from 0.1 to 0.9. For the last three cases, the parameters except the varied ones have the same settings as for the first case. The results of the experiments are presented in the following subsections.

For verifying the effectiveness and efficiency of GP4ESP, we choose 11 following classical and up-to-date meta-heuristic algorithms for performance comparison in solving ESP. The parameters set for these meta-heuristic algorithms are shown in Table 4.

- **GA** (Genetic Algorithm) evolves the population based on Darwin's theory of evolution by crossover, mutation and selection operators, which is one of the most widely used and representative meta-heuristic algorithms (*Katoch, Chauhan & Kumar, 2021*). For each individual in every evolutionary iteration time, GA crosses it with another individual selected by the selection operator and mutating it.

- **DE** (Differential Evolution) integrates different between individuals into evolutionary operators (*Mohamed, Hadi & Jambi, 2019*). The difference of DE to GA mainly is that every offspring generated by the mutation is based on the offset calculated by randomly selected three individuals.

- **SA** (Simulated Annealing) (*Mahjoubi, Grinnemo & Taheri, 2022*) is a single-based solution, which searches with depth-first search idea and the heuristic idea simulating the physical cooling phenomenon. The evolution of SA consists of two-level nested loop, where the outer loop is cooling and the inner one is searching the best in the current temperature. The cooling is decreasing the current temperature by multiplying it with a defined attenuation coefficient ($< 1$). The inner loop repeats updating the current solution by Metropolis principle.

- **PSO** (particle swarm optimization) (*Nayak et al., 2023*) is simulating the birds flock's looking for food for population updating. PSO updates the velocity based on the current velocity (inertia), the personal best (self-cognition), and the global best (social-cognition) for each individual.

- **ABC** (Artificial Bee Colony) (*Zhou, Lu & Zhang, 2023*) mimics the honey bee swarms' foraging for food. In each iterative time, ABC is composed of three phases, employed bees, onlooker bees and scout bees for balancing the exploitation and exploration. The first phase generates new individual based on the difference between an individual

and randomly selected one. The second phase is to share information (fitness) among individuals. The last phase is dismissing and re-initiating employed bees with on changed fitness several times to explore new areas.

- **WOA** (Whale Optimization Algorithm) (*Mirjalili & Lewis, 2016*) mimics the social behaviour of humpback whales' hunting, which also has three phases of encircling prey, bubble-net attacking (exploitation) and search for prey(exploration) in searching. The encircling phase is moving toward the global best(prey). In the exploitation phase, humpback whales swim around prey in an ever-shrinking circle and following a spiral path. In the exploration phase, randomly selected humpback whales expand their searching areas.
- **HHO** (Harris Hawks Optimization) (*Heidari et al., 2019*) mimics the social behaviour of Harris' Hawks for hunting. The exploitation phase of HHO is using soft surrounding strategy, hard surrounding strategy and soft surrounding strategy with approaching fast dive, randomly, for updating positions of Harris' Hawks. The exploration phase is to move toward between the best (rabbit position) and the midpoint of Harris' Hawks.
- **GWO** (Grey Wolf Optimization) (*Mirjalili, Mirjalili & Lewis, 2014*) are mimicking the social behaviour of grey wolfs' hunting. GWO sets the first three best as $\alpha$, $\beta$, $\gamma$ wolves, and others as $\delta$ wolves. Then, GWO updates wolves' positions for prey hunting (searching), guided by $\alpha$, $\beta$, $\gamma$ wolves.
- **PSOM** (PSO with mutation operator) (*Hafsi, Gharsellaoui & Bouamama, 2022*) is one of recent hybrid meta-heuristic algorithm. It performs a mutation operator on each particle in each iteration to improve the population diversity.
- **PSOSA** (PSO and SA) (*Lin et al., 2023*) is a hybrid meta-heuristic algorithm proposed recently. It conducts SA operator on each particle once in each iteration of PSO, which updates the positions and accepts the update with Metropolis principle.
- **nPSO** (niching PSO) (*Zhang et al., 2023*) employs niching technology in PSO to solve the multimodal ESP problem. This method divides particles into several niches based on their position similarity, and replaces the global best with the niching best for updating particles' velocities.

For each group of experiments that evaluate the performance of all 12 algorithms, we repeat 11 times. We use the overall average response time as the performance metric. We also compare the time consumption of these algorithms in solving ESP problems in the follows. All algorithms are implemented by Python, and is opened in GitHub (https://github.com/wangXJTU/GP4ESP). The experiments are conducted in the computer with Windows 11 Home Basic, Intel i7-14700(F) processor, RTX4060Ti graphics card with 8GB GDDR6, 16GB DDR5 5600 RAM, and 1TB PCle 4.0 SSD. The python version is 3.11.7, and Numpy 1.26.4 is used for operating on matrices and vectors.

## Performance comparison

Figure 6 shows the overall average response time achieved by different algorithms in solving ESP in the first case. From the figure, GP4ESP achieve about 18.2%–20.7% shorter response time than others, which verifies the high efficiency of GP4ESP for ESP solutions. The main benefit of GP4ESP is effectively fusing the advantages of GA and PSO. GP4ESP has 18.3%

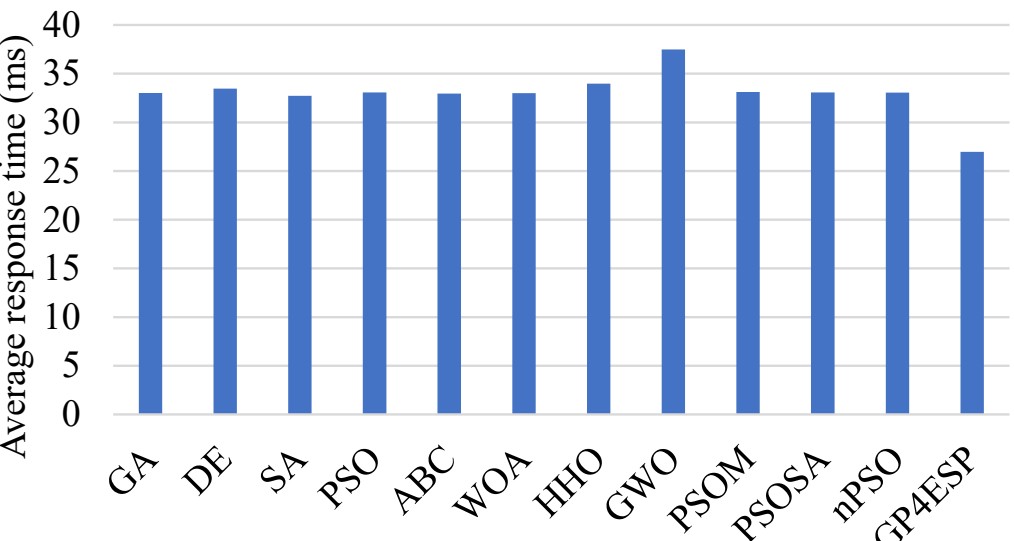

**Figure 6** **The average response time achieved by various ESP methods.**

and 18.5% better performance than GA and PSO, respectively, as shown in Fig. 6, which confirms the effectiveness of the fusion scheme exploited by GP4ESP for designing hybrid meta-heuristic algorithms. In addition, PSOM is also a hybrid PSO and GA algorithm, but has poorer performance, compared with GP4ESP. Besides, compared with PSOSA, another hybrid meta-heuristic algorithm, GP4ESP achieves 18.5% better average response time. These results substantiate the high efficiency of GP4ESP's fusion scheme. In the Table 5, we present the numerical results achieved by the first group of experiment that is repeated 11 times. In this table, we can see that GP4ESP always achieves the best performance in improving response time. This confirms the superior performance and the stability of GP4ESP.

The performance superior of GP4ESP is mainly coming from the hybridization approach that integrates the swam intelligence of PSO into the evolutionary process of GA. This not only improves the exploitation power of GA by the self- and social-cognition to ensure the convergence, but also guarantees the population diversity by the operators of GA to ensure the exploration ability. Experiment results verify that GP4ESP achieves a good balance between exploration and exploitation for searching ESP solutions, where GP4ESP have better performance than not only GA and PSO but also other hybrid meta-heuristic algorithms (PSOM and PSOSA).

Figure 7 gives the average time consumption of various algorithms. As shown in the figure, GP4ESP requires more time for solving ESP, compared with other algorithms excluding SA. This is because GP4ESP performs two more crossover operators on each individual in every iteration than GA, and GA has comparable time consumption with other algorithms. For SA, it conducts a depth-first search scheme on an individual, which can be very time-consuming, especially in solving large scale optimization problems. Noticing that ESP is not time-sensitive, it would be better to achieve a more efficient solution with

**Table 5  The response time and orders of various algorithms in conducted "Fixed" experiment 11 times.**

| Time | | GA | DE | SA | PSO | ABC | WOA | HHO | GWO | PSOM | PSOSA | nPSO | GP4ESP |
|------|------|------|------|------|------|------|------|------|------|------|------|------|------|
| 1 | Delay | 32.53 | 33.33 | 32.72 | 32.96 | 33.14 | 33.36 | 34.87 | 36.67 | 32.99 | 33.02 | 32.72 | 26.70 |
| | Rank | 2 | 9 | 3 | 5 | 8 | 10 | 11 | 12 | 6 | 7 | 4 | 1 |
| 2 | Delay | 33.45 | 34.48 | 33.20 | 33.48 | 33.12 | 33.43 | 34.00 | 37.37 | 33.44 | 33.39 | 33.62 | 27.93 |
| | Rank | 7 | 11 | 3 | 8 | 2 | 5 | 10 | 12 | 6 | 4 | 9 | 1 |
| 3 | Delay | 33.10 | 34.48 | 32.92 | 33.05 | 33.29 | 33.97 | 34.76 | 38.09 | 33.38 | 33.21 | 33.36 | 27.92 |
| | Rank | 4 | 10 | 2 | 3 | 6 | 9 | 11 | 12 | 8 | 5 | 7 | 1 |
| 4 | Delay | 32.88 | 33.33 | 33.05 | 33.10 | 33.00 | 33.40 | 33.90 | 37.34 | 33.32 | 33.24 | 33.22 | 26.42 |
| | Rank | 2 | 9 | 4 | 5 | 3 | 10 | 11 | 12 | 8 | 7 | 6 | 1 |
| 5 | Delay | 32.64 | 33.33 | 32.51 | 32.62 | 32.55 | 32.65 | 33.48 | 37.33 | 32.85 | 32.53 | 32.75 | 26.20 |
| | Rank | 6 | 10 | 2 | 5 | 4 | 7 | 11 | 12 | 9 | 3 | 8 | 1 |
| 6 | Delay | 33.17 | 33.33 | 32.70 | 33.28 | 32.97 | 32.96 | 34.19 | 37.20 | 33.20 | 33.12 | 33.11 | 26.67 |
| | Rank | 7 | 10 | 2 | 9 | 4 | 3 | 11 | 12 | 8 | 6 | 5 | 1 |
| 7 | Delay | 33.02 | 33.33 | 32.90 | 33.28 | 32.91 | 32.68 | 33.54 | 38.00 | 33.19 | 33.26 | 33.24 | 27.13 |
| | Rank | 5 | 10 | 3 | 9 | 4 | 2 | 11 | 12 | 6 | 8 | 7 | 1 |
| 8 | Delay | 33.15 | 33.33 | 32.48 | 33.17 | 33.24 | 33.42 | 35.00 | 37.87 | 33.28 | 33.24 | 33.11 | 26.95 |
| | Rank | 4 | 9 | 2 | 5 | 6 | 10 | 11 | 12 | 8 | 7 | 3 | 1 |
| 9 | Delay | 32.17 | 32.26 | 31.82 | 32.23 | 32.18 | 31.53 | 33.32 | 36.98 | 32.04 | 32.17 | 32.15 | 25.78 |
| | Rank | 7 | 10 | 3 | 9 | 8 | 2 | 11 | 12 | 4 | 6 | 5 | 1 |
| 10 | Delay | 33.36 | 33.33 | 33.00 | 33.41 | 33.44 | 33.26 | 34.07 | 37.01 | 33.32 | 33.43 | 33.54 | 27.50 |
| | Rank | 6 | 5 | 2 | 7 | 9 | 3 | 11 | 12 | 4 | 8 | 10 | 1 |
| 11 | Delay | 33.12 | 33.33 | 32.69 | 33.19 | 32.85 | 32.66 | 33.53 | 37.60 | 33.09 | 33.24 | 32.54 | 27.08 |
| | Rank | 7 | 10 | 4 | 8 | 5 | 3 | 11 | 12 | 6 | 9 | 2 | 1 |
| Average | Delay | 33.01 | 33.46 | 32.73 | 33.08 | 32.96 | 33.00 | 33.98 | 37.48 | 33.11 | 33.08 | 33.06 | 26.96 |
| | Rank | 5 | 10 | 2 | 7 | 3 | 4 | 11 | 12 | 9 | 8 | 6 | 1 |

some time overhead. Besides, GP4ESP requires only less than 1.5 min for solving ESP with 1,000 edge stations and 600 ESs. These confirm the practicability of GP4ESP in planning edge computing architecture solutions.

We also perform $t$-tests to show that The performance differences between GP4ESP and other algorithms are statistically significant. Table 6 gives the $p$-values achieved by $t$-tests. As shown in the table, all $p$-values are lower than 0.01, which confirms GP4ESP has statistically significant better overall average response time than other algorithms, verifying the high efficiency of GP4ESP again.

## Performance varied with system parameters

Figures 8–10 present the average response time achieved by various algorithms for solving ESP with varied system scale, ES number and load, respectively. As shown in Fig. 8, as the problem scale is increased, the average response time is stable for all algorithms. This is because the improvement opportunity on the response time is mainly related to the load (the ratio of arrival rate to service rate) according to queuing theory and the load is (almost) not changed with the problem scale. In Fig. 9, we can see that the response time is decreased linearly with the ES number, which is mainly because the service capacity is

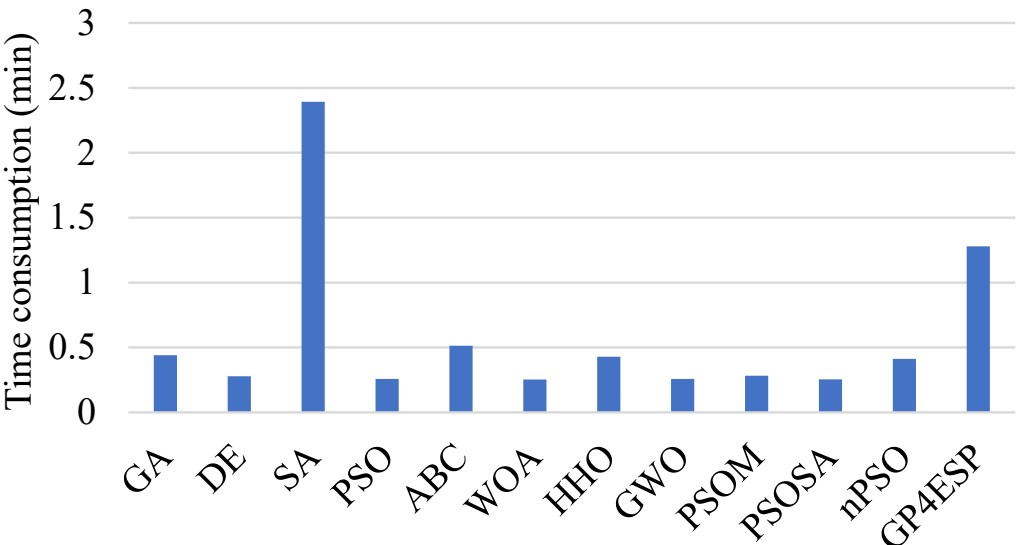

**Figure 7** **The time consumed by various ESP methods for solving ESP.**

**Table 6** *p*-values achieved by *t*-test on whether the performance of GP4ESP is different from other algorithms.

|  | Average response time | Time consumption |
| --- | --- | --- |
| GA | $1.47 \times 10^{-14}$ | $5.26 \times 10^{-15}$ |
| DE | $2.77 \times 10^{-16}$ | $9.01 \times 10^{-21}$ |
| SA | $3.28 \times 10^{-14}$ | $2.43 \times 10^{-11}$ |
| PSO | $1.75 \times 10^{-14}$ | $1.61 \times 10^{-16}$ |
| ABC | $2.79 \times 10^{-14}$ | $5.13 \times 10^{-15}$ |
| WOA | $1.33 \times 10^{-15}$ | $1.02 \times 10^{-15}$ |
| HHO | $4.70 \times 10^{-17}$ | $5.49 \times 10^{-15}$ |
| GWO | $2.61 \times 10^{-19}$ | $7.65 \times 10^{-16}$ |
| PSOM | $7.60 \times 10^{-15}$ | $3.45 \times 10^{-16}$ |
| PSOSA | $9.41 \times 10^{-15}$ | $1.32 \times 10^{-15}$ |
| nPSO | $2.79 \times 10^{-15}$ | $3.57 \times 10^{-15}$ |

increased with the ES number. The response time is increased linearly with the load, shown in Fig. 10, which is consistent with queuing theory. In all of these cases, GP4ESP always achieves the best response time, proving the efficiency of GP4ESP further and its general applicability.

For all of these meta-heuristic algorithms, their time complexities are linearly increased with the population size, the iterative times, the dimension of each individual (*i.e.,* number of stations) and the number of ES, due to following reasons. Meta-heuristic algorithms perform various updating operators on each individual at all dimensions in every iteration of population updating. Besides, for evaluating the fitness for each individual in every iteration, the response time of every edge station needs to be calculated based on the

**Peer**J Computer Science

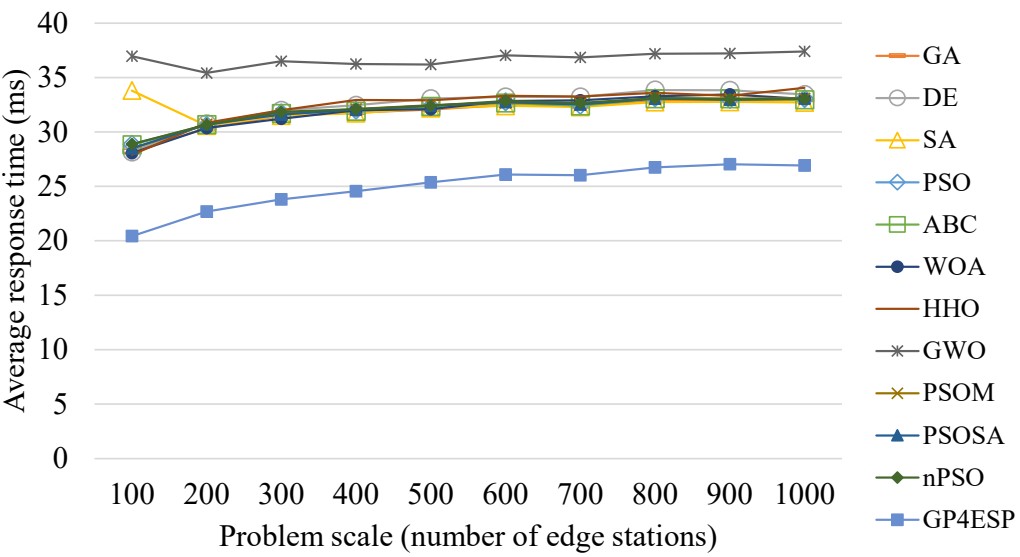

**Figure 8** The average response time achieved by various ESP methods with varied problem scale.

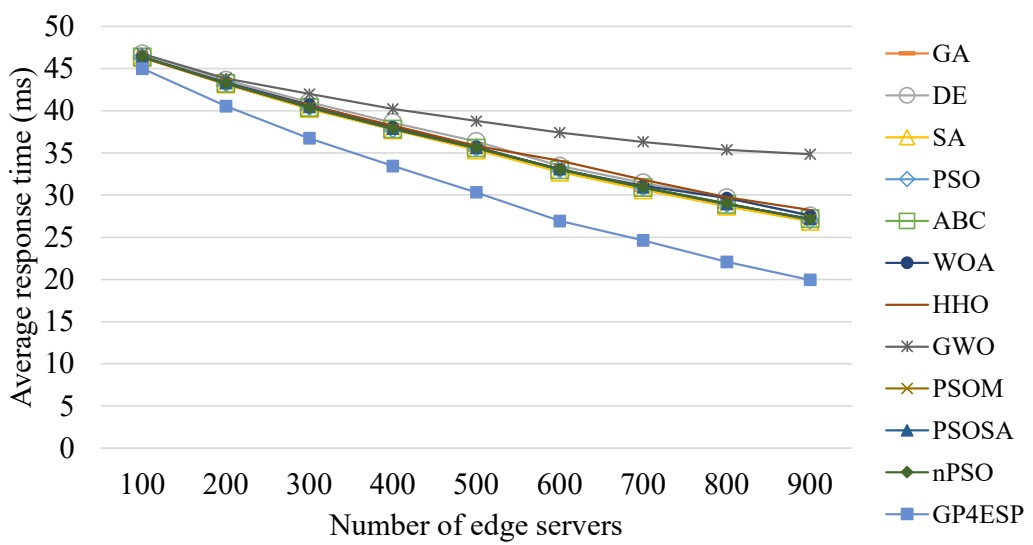

**Figure 9** The average response time achieved by various ESP methods with varied number of ESs.

corresponding ESP solution. The calculation of the response time of edge stations that ESs
are placed in has more time consumption than that of stations that ESs are not placed in,
as the former performs Eq. (5) while the latter only sets the response time as the cloud's
processing delay. Therefore, all algorithms consume time linearly increased with the
problem scale and the ES number, as shown in Figs. 11 and 12, respectively, and relatively
unchanged with the varied load, as shown in Fig. 13. These experiment results verify the
good scalability of meta-heuristic algorithms.

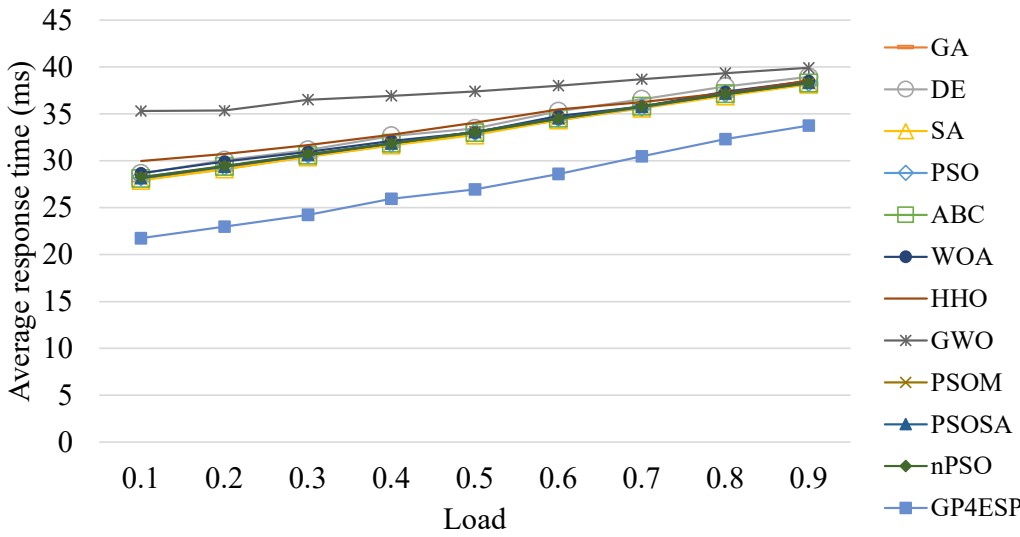

**Figure 10** **The average response time achieved by various ESP methods with varied overall load.**

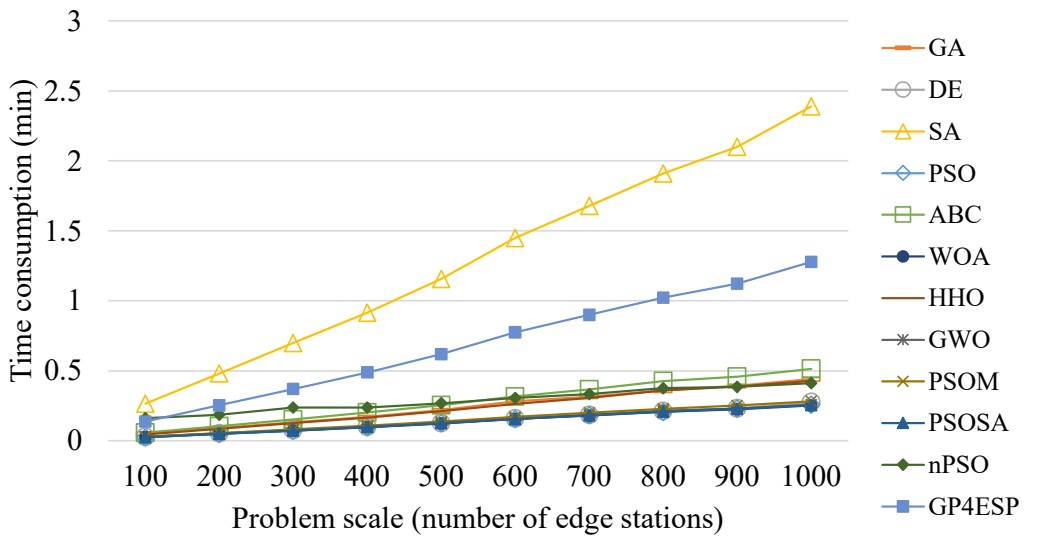

**Figure 11** **The time consumptions of various ESP methods with varied problem scale.**

## CONCLUSION

In this article, edge server placement is studied, aiming at making the best decision on the position where each purchased edge server by the service provider to build or upgrade its edge computing solution. First, ESP problem is formulated into a binary optimization problem with the objective of minimizing the overall average response time of all requests. Then, for solving the ESP problem with a reasonable time for large scale systems, a hybrid meta-heuristic algorithm is proposed by combining both advanced population evolution ideas of GA and PSO. At last, extensive experiments are conducted and results verified the

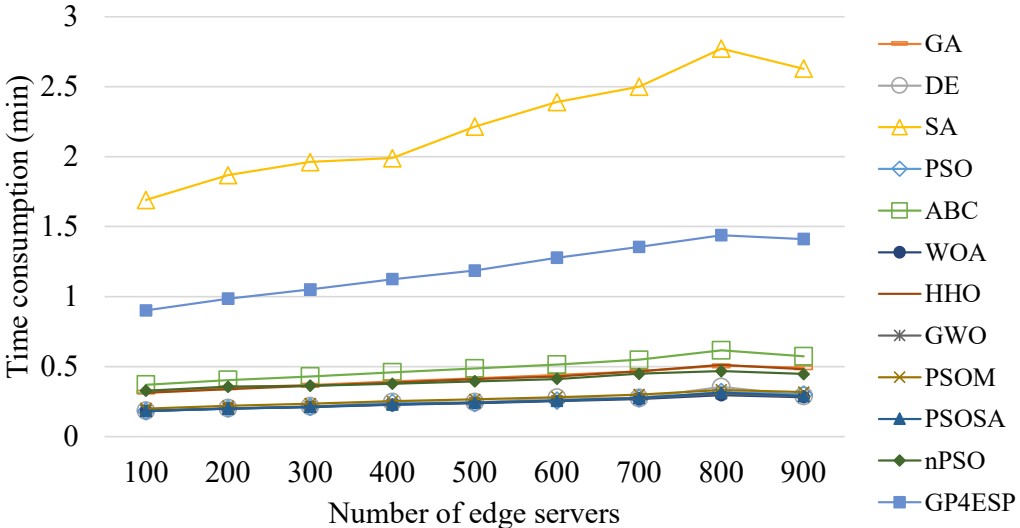

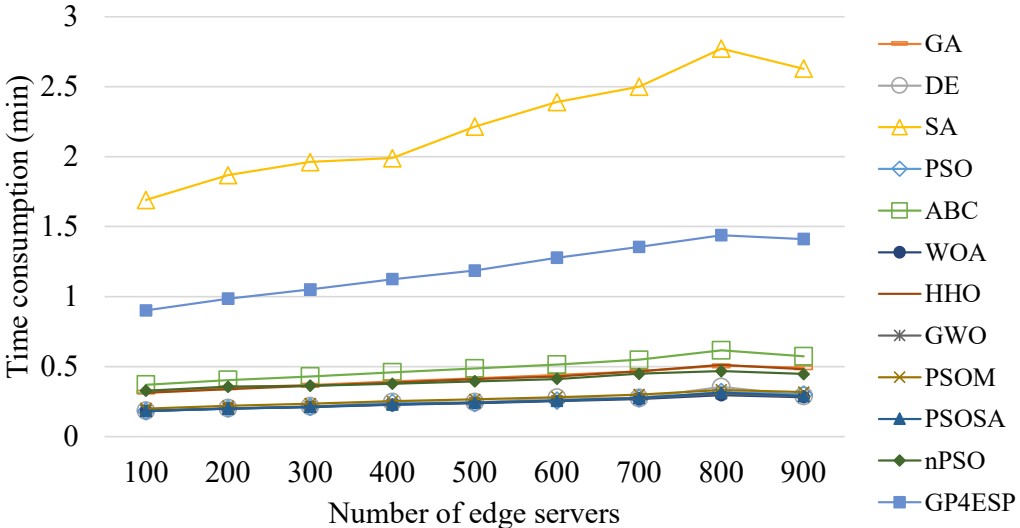

**Figure 12** The time consumptions of various ESP methods with varied number of ESs.

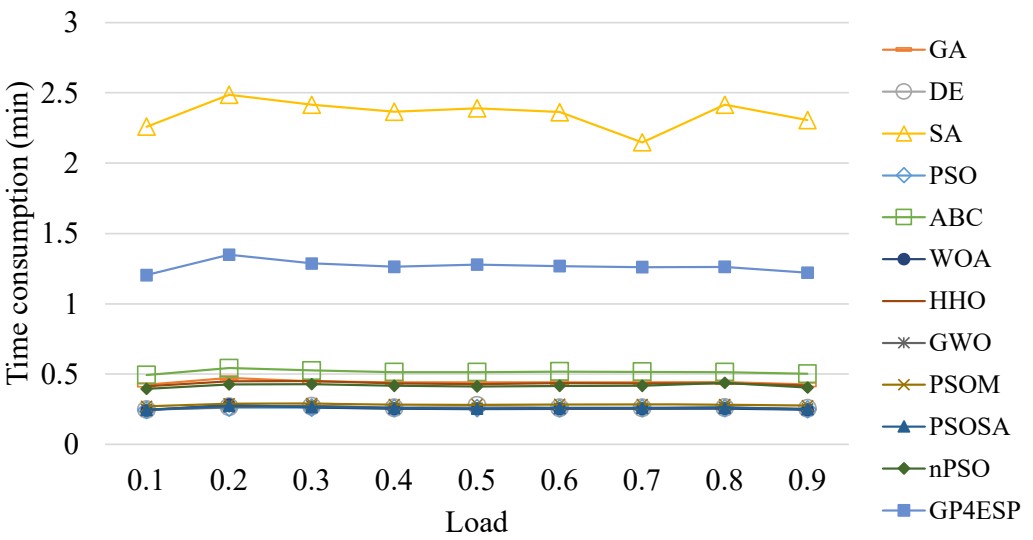

**Figure 13** The time consumptions of various ESP methods with varied overall load.

performance superiority of the proposed hybrid meta-heuristic algorithm in solving ESP problems.

In this article, we consider a relatively simple edge computing environment without considering the cooperation between different edge stations by redistributing some requests from one station with high load to another one with adequate edge resources. In the future,

we will expand our hybridization idea for more hybrid meta-heuristic algorithms and supporting more edge computing scenarios.

### Funding

The research was supported by the National Natural Science Foundation of China (Grant No. 62102372), the key scientific and technological projects of Henan Province (Grant No. 232102210078 and 242102210095), the Natural Science Foundation of Henan (Grant No. 222300420582), the Doctor Scientific Research Fund of Zhengzhou University of Light Industry (Grant No. 2021BSJJ029) and the Key Scientific Research Project of Higher Education Institutions in Henan Province (Grant No. 24B413005). The funders had no role in study design, data collection and analysis, decision to publish, or preparation of the manuscript.

### Grant Disclosures

The following grant information was disclosed by the authors:
National Natural Science Foundation of China: No. 62102372.
The key scientific and technological projects of Henan Province: No. 232102210078, 242102210095.
Natural Science Foundation of Henan: No. 222300420582.
The Doctor Scientific Research Fund of Zhengzhou University of Light Industry: No. 2021BSJJ029.
Key Scientific Research Project of Higher Education Institutions in Henan Province: No. 24B413005.

### Competing Interests

The authors declare there are no competing interests.

### Author Contributions

- Fang Han performed the experiments, analyzed the data, prepared figures and/or tables, and approved the final draft.
- Hui Fu performed the experiments, analyzed the data, authored or reviewed drafts of the article, and approved the final draft.
- Bo Wang conceived and designed the experiments, analyzed the data, performed the computation work, prepared figures and/or tables, authored or reviewed drafts of the article, and approved the final draft.
- Yaoli Xu performed the experiments, performed the computation work, prepared figures and/or tables, authored or reviewed drafts of the article, and approved the final draft.
- Bin Lv conceived and designed the experiments, performed the computation work, authored or reviewed drafts of the article, and approved the final draft.

### Data Availability

The source codes and raw data are available in the Supplemental Files.

## Supplemental Information

Supplemental information for this article can be found online at http://dx.doi.org/10.7717/peerj-cs.2439#supplemental-information.

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
