# Peer review of "GP4ESP: a hybrid genetic algorithm and particle swarm optimization algorithm for edge server placement"

_PeerJ Computer Science, doi:10.7717/peerj-cs.2439_

## Round 0.1 · original submission · Major Revisions

Dear authors,

The reviews for your manuscript are included at the bottom of this letter. We ask that you make necessary changes and additions to your manuscript based on those concerns and criticisms. Furthermore, adding a discussion for synthesis of findings, implications, future research, and limitations will be better. When submitting the revised better, followings should also be addressed:

1. Abstract does not clearly explain the contribution. The motivation of the paper does not exist. The contribution is not properly explained in an understandable way. The abstract section should be rewritten in order to clearly state the manuscript's main focus. The abstract should give the readers essential details, i.e., including the main contributions, the proposed method, the main problem, the obtained results, the benchmark tests, the comparative methods, etc. Efforts are needed to make the abstract coherent while clearly describing the problem being investigated and findings.
2. The reason for selecting the particle swarm optimization algorithm and evolutionary strategy algorithm for hybridization is not discussed.
3. The current introduction is very simple and misses many contents related to the problem formulation. There is not a clear categorization of related work. Introduction section seems broad, voluminous, and heterogeneous The authors are supposed to focus on the main topic of the study and present a Literature Review in the form of tables in order to make research gaps and innovations easy to detect. Authoritative synthesis assessing the current state-of-the-art is absent.
4. Configuration space of evolutionary algorithms should be detailed. It should be more specific and comprehensive. Representation scheme (encoding type) and fitness function with constraint functions should be clearly provided
5. How constraints (for decision variables and constraint functions) are handled is not clear.
6. The research gaps and contributions should be clearly summarized in the introduction section. Please evaluate how your study is different from others.
7. The values for the parameters of the algorithms selected for comparison should be given.
8. There is no detailed discussion and the authors do not explain why the proposed method is superior.
9. The paper lacks the running environment, including software and hardware. The analysis and configurations of experiments should be presented in detail for reproducibility.

Best wishes,

Reviewer 1 ·

Basic reporting

1. Some equations need refinement. For example equation 2 in 'Problem Formulaion' section.
2. The 'Related Work' section can be moved up.
3. The abstract should clearly contain a brief statement on the achieved result.
4. The introduction section can be expanded to include some background on the ESP problem.
5. The manuscript should be rechecked thoroughly for grammatical errors.

Experimental design

1. The flowchart for the process is given but the manuscript lacks a clear demonstration of the algorithm. The step should be clearly stated and demonstrated.
2. Supplement the proposed algorithm with a pseudo-code and include additional figures for assistance.
3. Include equations to support the steps of the algorithm.
4. Clearly explain the parameters used in the experiments.
5. Table 4 can be re-written after formatting the time-values. The current format is difficult to comprehend.
6. A brief paragraph on the mechanisms used by the benchmark algorithms (like GA, SA,etc.) to solve this problem would be beneficial for an improved understanding.

Validity of the findings

1. The experiments justify the proposed algorithm and overall aim of the study.

Reviewer 2 ·

Basic reporting

The paper is well written, but it has some typos.

The paper does not show the benefits of combing GA with PSO. Why not using other optimizers instead of PSO such as WOA.

In some places, the format of citation should be changed. For example,
line 32: Dayong et al. (2024) should be changed to (Dayong et al. (2024)).
Line 269 Zhou et al. Zhou et al. (2023) should be changed to Zhou et al. (2023)

Related work is not enough. Some related works are not mentioned. For example:
Edge Server Placement for Service Offloading in Internet of Things. This paper is based on GA and PSO similar this paper.
Another related paper is: An Improved Whale Optimization Algorithm for Optimal Placement of Edge Server, where the paper shows that their methods is better than PSO.

Experimental design

More elaboration on the proposed algorithm is required especially on the GA algorithm such as crossover, the mutation, population size, and the number of iterations.

The way of combing GA and PSO should be explained in more details.

One of the contributions of the paper is formulating ESP into a binary linear programming problem in a universal three-layer device-edge cloud computing framework. How this formulation is different than the ones presented in the literature.

Validity of the findings

The experimental results are clear, but it would be better to compare with more recent works instead of classical optimizers.

---

## Round 0.2 · accepted · Accept

Dear authors,

Thank you for the revised paper. The reviewers think that you have performed the necessary additions and modifications. Your paper now seems sufficiently improved and acceptable for publication. In production step, please fix a few minor typos specified by Reviewer 1.

Best wishes,

Reviewer 1 ·

Basic reporting

1. The writing has been improved, but there are still some ways this can be enhanced further- for example - rewrite line 68, 91.
2. In line 88, it should be 'fourth' instead of 'forth'.
3. In Fig 2, in the last step 'decoding' should be replaced with 'Decoding'.
4. In the pseudo-code of the algorithm 'Algorithm 1', the actions should be written in present tense instead of present continuos tense. For example - 'initializing' should be written as 'initialize', 'evaluating' should be written as 'evaluate', etc.

Experimental design

No further comments.

Validity of the findings

No further comments.

Additional comments

-

Reviewer 2 ·

Basic reporting

no comments

Experimental design

no comments

Validity of the findings

no comments

Additional comments

no comments